# Inverse Reward Design

**Dylan Hadfield-Menell**     **Smitha Milli**     **Pieter Abbeel**[*]     **Stuart Russell**     **Anca D. Dragan**
Department of Electrical Engineering and Computer Science
University of California, Berkeley
Berkeley, CA 94709
{dhm, smilli, pabbeel, russell, anca}@cs.berkeley.edu

## Abstract

Autonomous agents optimize the reward function we give them. What they don't know is how hard it is for us to design a reward function that actually captures what we want. When designing the reward, we might think of some specific training scenarios, and make sure that the reward will lead to the right behavior in *those* scenarios. Inevitably, agents encounter *new* scenarios (e.g., new types of terrain) where optimizing that same reward may lead to undesired behavior. Our insight is that reward functions are merely *observations* about what the designer *actually* wants, and that they should be interpreted in the context in which they were designed. We introduce *inverse reward design* (IRD) as the problem of inferring the true objective based on the designed reward and the training MDP. We introduce approximate methods for solving IRD problems, and use their solution to plan risk-averse behavior in test MDPs. Empirical results suggest that this approach can help alleviate negative side effects of misspecified reward functions and mitigate reward hacking.

## 1 Introduction

Robots[2] are becoming more capable of optimizing their reward functions. But along with that comes the burden of making sure we specify these reward functions correctly. Unfortunately, this is a notoriously difficult task. Consider the example from Figure 1. Alice, an AI engineer, wants to build a robot, we'll call it Rob, for mobile navigation. She wants it to reliably navigate to a target location and expects it to primarily encounter grass lawns and dirt pathways. She trains a perception system to identify each of these terrain types and then uses this to define a reward function that incentivizes moving towards the target quickly, avoiding grass where possible. When Rob is deployed into the world, it encounters a novel terrain type; for dramatic effect, we'll suppose that it is lava. The terrain prediction goes haywire on this out-of-distribution input and generates a meaningless classification which, in turn, produces an arbitrary reward evaluation. As a result, Rob might then drive to its demise. This failure occurs because the reward function Alice *specified* implicitly through the terrain predictors, which ends up outputting arbitrary values for lava, is *different* from the one Alice *intended*, which would actually penalize traversing lava.

In the terminology from Amodei et al. (2016), this is a *negative side effect* of a *misspecified reward* — a failure mode of reward design where leaving out important aspects leads to poor behavior. Examples date back to King Midas, who wished that everything he touched turn to gold, leaving out that he didn't mean his food or family. Another failure mode is *reward hacking*, which happens when, e.g., a vacuum cleaner ejects collected dust so that it can collect even more (Russell & Norvig, 2010), or a racing boat in a game loops in place to collect points instead of actually winning the race (Amodei & Clark, 2016). Short of requiring that the reward designer anticipate and penalize all possible misbehavior in advance, how can we alleviate the impact of such reward misspecification?

---

[*]OpenAI, International Computer Science Institute (ICSI)
[2]Throughout this paper, we will use robot to refer generically to any artificial agent.

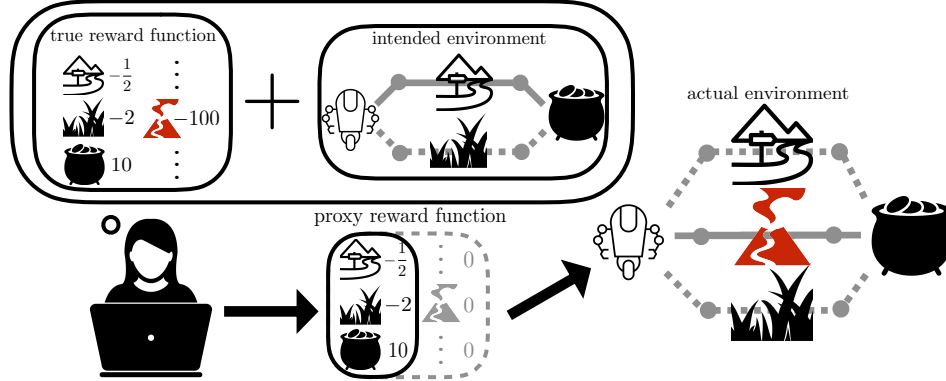

Figure 1: An illustration of a *negative side effect*. Alice designs a reward function so that her robot navigates to the pot of gold and prefers dirt paths. She does not consider that her robot might encounter lava in the real world and leaves that out of her reward specification. The robot maximizing this *proxy* reward function drives through the lava to its demise. In this work, we formalize the (Bayesian) *inverse reward design* (IRD) problem as the problem of inferring (a distribution on) the *true* reward function from the proxy. We show that IRD can help mitigate unintended consequences from misspecified reward functions like negative side effects and reward hacking.

*We leverage a key insight: that the designed reward function should merely be an observation about the intended reward, rather than the definition; and should be interpreted in the context in which it was designed.* First, a robot should have uncertainty about its reward function, instead of treating it as fixed. This enables it to, e.g., be risk-averse when planning in scenarios where it is not clear what the right answer is, or to ask for help. Being uncertain about the true reward, however, is only half the battle. To be effective, a robot must acquire the *right* kind of uncertainty, i.e. know what it knows and what it doesn't. We propose that the 'correct' shape of this uncertainty depends on the environment for which the reward was designed.

In Alice's case, the situations where she tested Rob's learning behavior did not contain lava. Thus, the lava-*avoiding* reward would have produced *the same behavior* as Alice's designed reward function in the (lava-free) environments that Alice considered. A robot that knows the settings it was evaluated in should also know that, even though the designer specified a lava-*agnostic* reward, they might have actually meant the lava-*avoiding* reward. Two reward functions that would produce similar behavior in the training environment should be treated as equally likely, regardless of which one the designer actually specified. We formalize this in a probabilistic model that relates the proxy (designed) reward to the true reward via the following assumption:

**Assumption 1.** *Proxy reward functions are likely to the extent that they lead to high **true** utility behavior in the **training** environment.*

Formally, we assume that the observed proxy reward function is the approximate solution to a *reward design problem* (Singh et al., 2010). Extracting the true reward is the *inverse* reward design problem.

The idea of using human behavior as observations about the reward function is far from new. Inverse reinforcement learning uses human demonstrations (Ng & Russell, 2000; Ziebart et al., 2008), shared autonomy uses human operator control signals (Javdani et al., 2015), preference-based reward learning uses answers to comparison queries (Jain et al., 2015), and even what the human wants (Hadfield-Menell et al., 2017). We observe that, *even when the human behavior is to actually write down a reward function*, this should still be treated as an observation, demanding its own observation model.

Our paper makes three contributions. First, we define the *inverse reward design* (IRD) problem as the problem of inferring the true reward function given a proxy reward function, an intended decision problem (e.g., an MDP), and a set of possible reward functions. Second, we propose a solution to IRD and justify how an intuitive algorithm which treats the proxy reward as a set of expert demonstrations can serve as an effective approximation. Third, we show that this inference approach, combined with risk-averse planning, leads to algorithms that are robust to misspecified rewards, alleviating both negative side effects as well as reward hacking. We build a system that 'knows-what-it-knows' about reward evaluations that automatically detects and avoids distributional shift in situations with high-dimensional features. Our approach substantially outperforms the baseline of literal reward interpretation.

## 2 Inverse Reward Design

**Definition 1.** *(Markov Decision Process Puterman (2009)) A (finite-horizon) Markov decision process (MDP), $M$, is a tuple $M = \langle \mathcal{S}, \mathcal{A}, T, r, H \rangle$. $\mathcal{S}$ is a set of states. $\mathcal{A}$ is a set of actions. $T$ is a probability distribution over the next state, given the previous state and action. We write this as $T(s_{t+1}|s_t, a)$. $r$ is a reward function that maps states to rewards $r : \mathcal{S} \mapsto \mathbb{R}$. $H \in \mathbb{Z}_+$ is the finite planning horizon for the agent.*

A solution to $M$ is a *policy*: a mapping from the current timestep and state to a distribution over actions. The optimal policy maximizes the expected sum of rewards. We will use $\xi$ to represent trajectories. In this work, we consider reward functions that are linear combinations of feature vectors $\phi(\xi)$. Thus, the reward for a trajectory, given weights $w$, is $r(\xi;\ w) = w^\top \phi(\xi)$.

The MDP formalism defines optimal behavior, given a reward function. However, it provides no information about where this reward function comes from (Singh et al., 2010). We refer to an MDP without rewards as a *world model*. In practice, a system designer needs to select a reward function that encapsulates the intended behavior. This process is *reward engineering* or *reward design*:

**Definition 2.** *(Reward Design Problem (Singh et al., 2010)) A reward design problem (RDP) is defined as a tuple $P = \langle r^*, \widetilde{M}, \widetilde{\mathcal{R}}, \pi(\cdot|\widetilde{r}, \widetilde{M}) \rangle$. $r^*$ is the true reward function. $\widetilde{M}$ is a world model. $\widetilde{\mathcal{R}}$ is a set of proxy reward functions. $\pi(\cdot|\widetilde{r}, \widetilde{M})$ is an agent model, that defines a distribution on trajectories given a (proxy) reward function and a world model.*

In an RDP, the designer believes that an agent, represented by the policy $\pi(\cdot|\widetilde{r}, \widetilde{M})$, will be deployed in $\widetilde{M}$. She must specify a *proxy reward function* $\widetilde{r} \in \widetilde{\mathcal{R}}$ for the agent. Her goal is to specify $\widetilde{r}$ so that $\pi(\cdot|\widetilde{r}, \widetilde{M})$ obtains high reward according to the true reward function $r^*$. We let $\widetilde{w}$ represent weights for the proxy reward function and $w^*$ represent weights for the true reward function.

In this work, our motivation is that system designers are fallible, so we should not expect that they perfectly solve the reward design problem. Instead we consider the case where the system designer is approximately optimal at solving a known RDP, which is distinct from the MDP that the robot currently finds itself in. By inverting the reward design process to infer (a distribution on) the true reward function $r^*$, the robot can understand where its reward evaluations have high variance and plan to avoid those states. We refer to this inference problem as the *inverse reward design* problem:

**Definition 3.** *(Inverse Reward Design) The inverse reward design (IRD) problem is defined by a tuple $\langle \mathcal{R}, \widetilde{M}, \widetilde{\mathcal{R}}, \pi(\cdot|\widetilde{r}, \widetilde{M}), \widetilde{r} \rangle$. $\mathcal{R}$ is a space of possible reward functions. $\widetilde{M}$ is a world model. $\langle -, \widetilde{M}, \widetilde{\mathcal{R}}, \pi(\cdot|\widetilde{r}, \widetilde{M}) \rangle$ partially specifies an RDP $P$, with an unobserved reward function $r^* \in \mathcal{R}$. $\widetilde{r} \in \widetilde{\mathcal{R}}$ is the observed proxy reward that is an (approximate) solution to $P$.*

In solving an IRD problem, the goal is to recover $r^*$. We will explore Bayesian approaches to IRD, so we will assume a prior distribution on $r^*$ and infer a posterior distribution on $r^*$ given $\widetilde{r}$ $P(r^*|\widetilde{r}, \widetilde{M})$.

## 3 Related Work

**Optimal reward design.** Singh et al. (2010) formalize and study the problem of designing optimal rewards. They consider a designer faced with a distribution of environments, a class of reward functions to give to an agent, and a *fitness* function. They observe that, in the case of bounded agents, it may be optimal to select a proxy reward that is distinct from the fitness function. Sorg et al. (2010) and subsequent work has studied the computational problem of selecting an optimal proxy reward.

In our work, we consider an alternative situation where the *system designer* is the bounded agent. In this case, the proxy reward function is distinct from the fitness function – the true utility function in our terminology – because system designers can make mistakes. IRD formalizes the problem of determining a true utility function given an observed proxy reward function. This enables us to design agents that are robust to misspecifications in their reward function.

**Inverse reinforcement learning.** In *inverse reinforcement learning* (IRL) (Ng & Russell, 2000; Ziebart et al., 2008; Evans et al., 2016; Syed & Schapire, 2007) the agent observes demonstrations of (approximately) optimal behavior and infers the reward function being optimized. IRD is a similar

problem, as both approaches infer an unobserved reward function. The difference is in the observation: IRL observes behavior, while IRD directly observes a reward function. Key to IRD is assuming that this observed reward incentivizes behavior that is approximately optimal with respect to the true reward. In Section 4.2, we show how ideas from IRL can be used to approximate IRD. Ultimately, we consider both IRD and IRL to be complementary strategies for *value alignment* (Hadfield-Menell et al., 2016): approaches that allow designers or users to communicate preferences or goals.

**Pragmatics.** The *pragmatic* interpretation of language is the interpretation of a phrase or utterance in the context of alternatives (Grice, 1975). For example, the utterance "some of the apples are red" is often interpreted to mean that "not all of the apples are red" although this is not literally implied. This is because, in context, we typically assume that a speaker who meant to say "all the apples are red" would simply say so.

Recent models of pragmatic language interpretation use two levels of Bayesian reasoning (Frank et al., 2009; Goodman & Lassiter, 2014). At the lowest level, there is a literal listener that interprets language according to a shared literal definition of words or utterances. Then, a speaker selects words in order to convey a particular meaning to the literal listener. To model pragmatic inference, we consider the probable meaning of a given utterance from this speaker. We can think of IRD as a model of pragmatic reward interpretation: the speaker in pragmatic interpretation of language is directly analogous to the reward designer in IRD.

## 4 Approximating the Inference over True Rewards

We solve IRD problems by formalizing Assumption 1: the idea that proxy reward functions are likely to the extent that they incentivize high utility behavior in the training MDP. This will give us a probabilistic model for how $\widetilde{w}$ is generated from the true $w^*$ and the training MDP $\widetilde{M}$. We will invert this probability model to compute a distribution $P(w = w^*|\widetilde{w}, \widetilde{M})$ on the true utility function.

### 4.1 Observation Model

Recall that $\pi(\xi|\widetilde{w}, \widetilde{M})$ is the designer's model of the probability that the robot will select trajectory $\xi$, given proxy reward $\widetilde{w}$. We will assume that $\pi(\xi|\widetilde{w}, \widetilde{M})$ is the maximum entropy trajectory distribution from Ziebart et al. (2008), i.e. the designer models the robot as approximately optimal: $\pi(\xi|\widetilde{w}, \widetilde{M}) \propto \exp(w^\top \phi(\xi))$. An optimal designer chooses $\widetilde{w}$ to maximize expected true value, i.e. $\mathbb{E}[w^{*\top}\phi(\xi)|\xi \sim \pi(\xi|\widetilde{w}, \widetilde{M})]$ is high. We model an approximately optimal designer:

$$P(\widetilde{w}|w^*, \widetilde{M}) \propto \exp\left(\beta \mathbb{E}\left[w^{*\top}\phi(\xi)|\xi \sim \pi(\xi|\widetilde{w}, \widetilde{M})\right]\right) \tag{1}$$

with $\beta$ controlling how close to optimal we assume the person to be. This is now a formal statement of Assumption 1. $w^*$ can be pulled out of the expectation, so we let $\widetilde{\phi} = \mathbb{E}[\phi(\xi)|\xi \sim \pi(\xi|\widetilde{w}, \widetilde{M})]$. Our goal is to invert (1) and sample from (or otherwise estimate) $P(w^*|\widetilde{w}, \widetilde{M}) \propto P(\widetilde{w}|w^*, \widetilde{M})P(w^*)$. The primary difficulty this entails is that we need to know the *normalized* probability $P(\widetilde{w}|w^*, \widetilde{M})$. This depends on its normalizing constant, $\widetilde{Z}(w)$, which integrates over possible proxy rewards.

$$P(w = w^*|\widetilde{w}, \widetilde{M}) \propto \frac{\exp\left(\beta w^\top \widetilde{\phi}\right)}{\widetilde{Z}(w)}P(w), \widetilde{Z}(w) = \int_{\widetilde{w}} \exp\left(\beta w^\top \widetilde{\phi}\right) d\widetilde{w}. \tag{2}$$

### 4.2 Efficient approximations to the IRD posterior

To compute $P(w = w^*|\widetilde{w}, \widetilde{M})$, we must compute $\widetilde{Z}$, which is intractable if $\widetilde{w}$ lies in an infinite or large finite set. Notice that computing the value of the integrand for $\widetilde{Z}$ is highly non-trivial as it involves solving a planning problem. This is an example of what is referred to as a *doubly-intractable* likelihood (Murray et al., 2006). We consider two methods to approximate this normalizing constant.

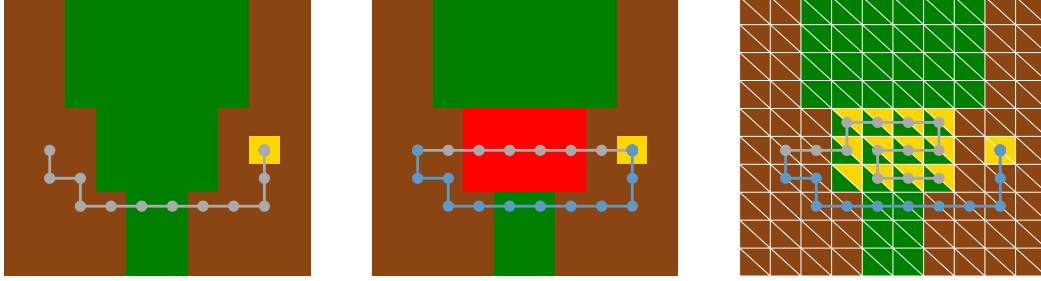

Figure 2: An example from the Lavaland domain. **Left:** The training MDP where the designer specifies a proxy reward function. This incentivizes movement toward targets (yellow) while preferring dirt (brown) to grass (green), and generates the gray trajectory. **Middle:** The testing MDP has lava (red). The proxy does not penalize lava, so optimizing it makes the agent go straight through (gray). This is a negative side effect, which the IRD agent avoids (blue): it treats the proxy as an observation in the context of the training MDP, which makes it realize that it cannot trust the (implicit) weight on lava. **Right:** The testing MDP has cells in which two sensor indicators no longer correlate: they look like grass to one sensor but target to the other. The proxy puts weight on the first, so the literal agent goes to these cells (gray). The IRD agent knows that it can't trust the distinction and goes to the target on which both sensors agree (blue).

**Sample to approximate the normalizing constant.** This approach, inspired by methods in approximate Bayesian computation (Sunnåker et al., 2013), samples a finite set of weights $\{w_i\}$ to approximate the integral in Equation 2. We found empirically that it helped to include the candidate sample $w$ in the sum. This leads to the normalizing constant

$$\hat{Z}(w) = \exp\left(w^\top \phi_w\right) + \sum_{i=0}^{N-1} \exp\left(\beta w^\top \phi_i\right). \tag{3}$$

Where $\phi_i$ and $\phi_w$ are the vector of feature counts realized optimizing $w_i$ and $w$ respectively.

**Bayesian inverse reinforcement learning.** During inference, the normalizing constant serves a calibration purpose: it computes how good the behavior produced by all proxy rewards in that MDP would be with respect to the true reward. Reward functions which increase the reward for *all* trajectories are not preferred in the inference. This creates an invariance to linear shifts in the feature encoding. If we were to change the MDP by shifting features by some vector $\phi_0$, $\phi \leftarrow \phi + \phi_0$, the posterior over $w$ would remain the same.

We can achieve a similar calibration and maintain the same property by directly integrating over the possible trajectories in the MDP:

$$Z(w) = \left(\int_\xi \exp(w^\top \phi(\xi))d\xi\right)^\beta ; \quad \hat{P}(w|\widetilde{w}) \propto \frac{\exp\left(\beta w^\top \widetilde{\phi}\right)}{Z(w)} \tag{4}$$

**Proposition 1.** *The posterior distribution that the IRD model induces on $w^*$ (i.e., Equation 2) and the posterior distribution induced by IRL (i.e., Equation 4) are invariant to linear translations of the features in the training MDP.*

*Proof.* See supplementary material. □

This choice of normalizing constant approximates the posterior to an IRD problem with the posterior from maximum entropy IRL (Ziebart et al., 2008). The result has an intuitive interpretation. The proxy $\widetilde{w}$ determines the average feature counts for a hypothetical dataset of expert demonstrations and $\beta$ determines the effective size of that dataset. The agent solves $\widetilde{M}$ with reward $\widetilde{w}$ and computes the corresponding feature expectations $\widetilde{\phi}$. The agent then pretends like it got $\beta$ demonstrations with features counts $\widetilde{\phi}$, and runs IRL. The more the robot believes the human is good at reward design, the more demonstrations it pretends to have gotten from the person. The fact that reducing the proxy to behavior in $\widetilde{M}$ approximates IRD is not surprising: the main point of IRD is that the proxy *reward* is merely a statement about what *behavior* is good in the training environment.

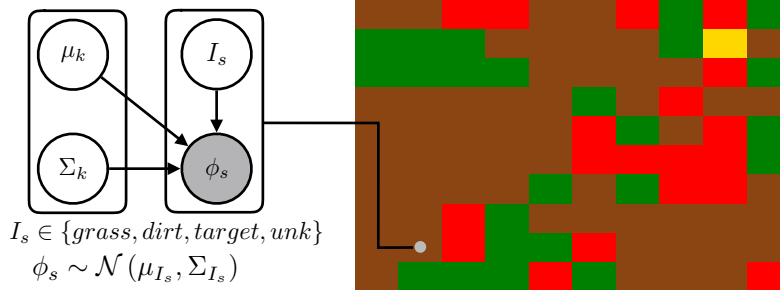

Figure 3: Our challenge domain with latent rewards. Each terrain type (grass, dirt, target, lava) induces a different distribution over high-dimensional features: $\phi_s \sim \mathcal{N}(\mu_{I_s}, \Sigma_{I_s})$. The designer never builds an indicator for lava, and yet the agent still needs to avoid it in the test MDPs.

## 5   Evaluation

### 5.1   Experimental Testbed

We evaluated our approaches in a model of the scenario from Figure 1 that we call *Lavaland*. Our system designer, Alice, is programming a mobile robot, Rob. We model this as a gridworld with movement in the four cardinal directions and four terrain types: target, grass, dirt, and lava. The true objective for Rob, $w^*$, encodes that it should get to the target quickly, stay off the grass, and avoid lava. Alice designs a proxy that performs well in a training MDP that does not contain lava. Then, we measure Rob's performance in a test MDP that *does* contain lava. Our results show that combining IRD and risk-averse planning creates incentives for Rob to avoid unforeseen scenarios.

We experiment with four variations of this environment: two proof-of-concept conditions in which the reward is misspecified, but the agent has direct access to feature indicators for the different categories (i.e. conveniently having a feature for lava); and two challenge conditions, in which the right features are *latent*; the reward designer does not build an indicator for lava, but by reasoning in the raw observation space and then using risk-averse planning, the IRD agent still avoids lava.

#### 5.1.1   Proof-of-Concept Domains

These domains contain feature indicators for the four categories: grass, dirt, target, and lava.

**Side effects in Lavaland.**   Alice expects Rob to encounter 3 types of terrain: grass, dirt, and target, and so she only considers the training MDP from Figure 2 (left). She provides a $\tilde{w}$ to encode a trade-off between path length and time spent on grass.

The training MDP contains no lava, but it is introduced when Rob is deployed. An agent that treats the proxy reward literally might go on the lava in the test MDP. However, an agent that runs IRD will know that it can't trust the weight on the lava indicator, since all such weights would produce the same behavior in the training MDP (Figure 2, middle).

**Reward Hacking in Lavaland.**   Reward hacking refers generally to reward functions that can be gamed or tricked. To model this within Lavaland, we use features that are correlated in the training domain but are uncorrelated in the testing environment. There are 6 features: three from one sensor and three from another sensor. In the training environment the features from both sensors are correct indicators of the state's terrain category (grass, dirt, target).

At test time, this correlation gets broken: lava looks like the target category to the second sensor, but the grass category to the first sensor. This is akin to how in a racing game (Amodei & Clark, 2016), winning and game points can be correlated at reward design time, but test environments might contain loopholes for maximizing points without winning. We want agents to hedge their bets between winning and points, or, in Lavaland, between the two sensors. An agent that treats the proxy reward function literally might go to these new cells if they are closer. In contrast, an agent that runs IRD will know that a reward function with the same weights put on the first sensor is just as likely as the proxy. Risk averse planning makes it go to the target for which both sensors agree (Figure 2, right).

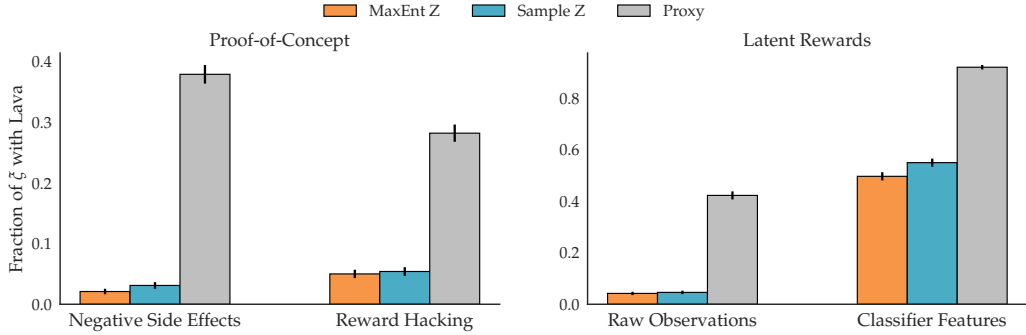

Figure 4: The results of our experiment comparing our proposed method to a baseline that directly plans with the proxy reward function. By solving an inverse reward design problem, we are able to create generic incentives to avoid unseen or novel states.

### 5.1.2 Challenge Domain: Latent Rewards, No More Feature Indicators

The previous examples allow us to explore reward hacking and negative side effects in an isolated experiment, but are unrealistic as they assume the existence of a feature indicator for unknown, unplanned-for terrain. To investigate misspecified objectives in a more realistic setting, we shift to the terrain type being latent, and inducing raw observations: we use a model where the terrain category determines the mean and variance of a multivariate Gaussian distribution over observed features. Figure 3 shows a depiction of this scenario. The designer has in mind a proxy reward on dirt, target, and grass, but *forgets that lava might exist*. We consider two realistic ways through which a designer might actually specify the proxy reward function, which is based on the terrain types that the robot does not have access to: 1) directly on the **raw observations** — collect samples of the training terrain types (dirt, grass, target) and train a (linear) reward predictor; or 2) **classifier features** — build a classifier to classify terrain as dirt, grass, or target, and define a proxy on its output.

Note that this domain allows for both negative side effects and reward hacking. Negative side effects can occur because the feature distribution for lava is different from the feature distribution for the three safe categories, and the proxy reward is trained only on the three safe categories. Thus in the testing MDP, the evaluation of the lava cells will be arbitrary so maximizing the proxy reward will likely lead the agent into lava. Reward hacking occurs when features that are correlated for the safe categories are uncorrelated for the lava category.

### 5.2 Experiment

**Lavaland Parameters.** We defined a distribution on map layouts with a log likelihood function that prefers maps where neighboring grid cells are the same. We mixed this log likelihood with a quadratic cost for deviating from a target ratio of grid cells to ensure similar levels of the lava feature in the testing MDPs. Our training MDP is 70% dirt and 30% grass. Our testing MDP is 5% lava, 66.5% dirt, and 28.5% grass.

In the proof-of-concept experiments, we selected the proxy reward function uniformly at random. For latent rewards, we picked a proxy reward function that evaluated to $+1$ for target, $+.1$ for dirt, and $-.2$ for grass. To define a proxy on raw observations, we sampled 1000 examples of grass, dirt, and target and did a linear regression. With classifier features, we simply used the target rewards as the weights on the classified features. We used 50 dimensions for our feature vectors. We selected trajectories via *risk-averse trajectory optimization*. Details of our planning method, and our approach and rationale in selecting it can be found in the supplementary material.

**IVs and DVs.** We measured the fraction of runs that encountered a lava cell on the test MDP as our dependent measure. This tells us the proportion of trajectories where the robot gets 'tricked' by the misspecified reward function; if a grid cell has never been seen then a conservative robot should plan to avoid it. We manipulate two factors: **literal-optimizer** and **Z-approx**. **literal-optimizer** is true if the robot interprets the proxy reward literally and false otherwise. **Z-approx** varies the approximation technique used to compute the IRD posterior. It varies across the two levels described in Section 4.2: sample to approximate the normalizing constant (**Sample-Z**) or use the normalizing constant from maximum entropy IRL (**MaxEnt-Z**) (Ziebart et al., 2008).

**Results.** Figure 4 compares the approaches. On the left, we see that IRD alleviates negative side effects (avoids the lava) and reward hacking (does not go as much on cells that look deceptively like the target to one of the sensors). This is important, in that the same inference method generalizes across different consequences of misspecified rewards. Figure 2 shows example behaviors.

In the more realistic latent reward setting, the IRD agent avoids the lava cells despite the designer forgetting to penalize it, and despite not even having an indicator for it: because lava is latent in the space, and so reward functions that would *implicitly* penalize lava are as likely as the one actually specified, risk-averse planning avoids it.

We also see a distinction between raw observations and classifier features. The first essentially matches the proof-of-concept results (note the different axes scales), while the latter is much more difficult across all methods. The proxy performs worse because each grid cell is classified before being evaluated, so there is a relatively good chance that at least one of the lava cells is misclassified as target. IRD performs worse because the behaviors considered in inference plan in the already classified terrain: a non-linear transformation of the features. The inference must both determine a good linear reward function to match the behavior *and* discover the corresponding uncertainty about it. When the proxy is a linear function of raw observations, the first job is considerably easier.

## 6 Discussion

**Summary.** In this work, we motivated and introduced the *Inverse Reward Design* problem as an approach to mitigate the risk from misspecified objectives. We introduced an observation model, identified the challenging inference problem this entails, and gave several simple approximation schemes. Finally, we showed how to use the solution to an inverse reward design problem to avoid side effects and reward hacking in a 2D navigation problem. We showed that we are able to avoid these issues reliably in simple problems where features are binary indicators of terrain type. Although this result is encouraging, in real problems we won't have convenient access to binary indicators for what matters. Thus, our challenge evaluation domain gave the robot access to only a high-dimensional observation space. The reward designer specified a reward based on this observation space which forgets to penalize a rare but catastrophic terrain. IRD inference still enabled the robot to understand that rewards which would implicitly penalize the catastrophic terrain are also likely.

**Limitations and future work.** IRD gives the robot a posterior distribution over reward functions, but much work remains in understanding how to best leverage this posterior. Risk-averse planning can work sometimes, but it has the limitation that the robot does not just avoid bad things like lava, it also avoids potentially good things, like a giant pot of gold. We anticipate that leveraging the IRD posterior for follow-up queries to the reward designer will be key to addressing misspecified objectives.

Another limitation stems from the complexity of the environments and reward functions considered here. The approaches we used in this work rely on explicitly solving a planning problem, and this is a bottleneck during inference. In future work, we plan to explore the use of different agent models that plan approximately or leverage, e.g., meta-learning (Duan et al., 2016) to scale IRD up to complex environments. Another key limitation is the use of linear reward functions. We cannot expect IRD to perform well unless the prior places weights on (a reasonable approximation to) the true reward function. If, e.g., we encoded terrain types as RGB values in Lavaland, there is unlikely to be a reward function in our hypothesis space that represents the true reward well.

Finally, this work considers one relatively simple error model for the designer. This encodes some implicit assumptions about the nature and likelihood of errors (e.g., IID errors). In future work, we plan to investigate more sophisticated error models that allow for systematic biased errors from the designer and perform human subject studies to empirically evaluate these models.

Overall, we are excited about the implications IRD has not only in the short term, but also about its contribution to the general study of the value alignment problem.

## Acknowledgements

This work was supported by the Center for Human Compatible AI and the Open Philanthropy Project, the Future of Life Institute, AFOSR, and NSF Graduate Research Fellowship Grant No. DGE 1106400.

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
