[Supplementary Material]

# Inverse Reward Design Supplementary Material

**Dylan Hadfield-Menell**    **Smitha Milli**    **Pieter Abbeel**[*]    **Stuart Russell**    **Anca D. Dragan**
Department of Electrical Engineering and Computer Science
University of California, Berkeley
Berkeley, CA 94709
{dhm, smilli, pabbeel, russell, anca}@cs.berkeley.edu

## 1   Proof of Proposition 1

### 1.1   Observation Model

Let $\pi(\xi|\widetilde{w}, \widetilde{M})$ be the probability that an agent optimizing $\widetilde{w}$ in $\widetilde{M}$ selects trajectory $\xi$. This is the designer's model of how the agent will use the designed reward. We will assume that $\pi(\xi|\widetilde{w}, \widetilde{M})$ is the maximum entropy trajectory distribution used in Ziebart et al. (2008), i.e. the designer models the robot as approximately optimal: $\pi(\xi|\widetilde{w}, \widetilde{M}) \propto \exp(w^\top \phi(\xi))$. An optimal designer chooses $\widetilde{w}$ to maximize expected true value, i.e. $\mathbb{E}[w^{*\top}\phi(\xi)|\xi \sim \pi(\xi|\widetilde{w}, \widetilde{M})]$ is high. We model an approximately optimal designer:

$$P(\widetilde{w}|w^*, \widetilde{M}) \propto \exp\left(\beta \mathbb{E}\left[w^{*\top}\phi(\xi)|\xi \sim \pi(\xi|\widetilde{w}, \widetilde{M})\right]\right) \tag{1}$$

with $\beta$ controlling how close to optimal we assume the person to be. $w^*$ can be pulled out of the expectation, so we let $\widetilde{\phi} = \mathbb{E}[\phi(\xi)|\xi \sim \pi(\xi|\widetilde{w}, \widetilde{M})]$.

Our goal is to invert (1) and sample from (or otherwise estimate) $P(w^*|\widetilde{w}, \widetilde{M}) \propto P(\widetilde{w}|w^*, \widetilde{M})P(w^*)$. The primary difficulty this entails is that we need to know the *normalized* probability $P(\widetilde{w}|w^*, \widetilde{M})$. This depends on its normalizing constant, $\widetilde{Z}(w)$, which integrates over possible proxy rewards.

$$P(w = w^*|\widetilde{w}, \widetilde{M}) \propto \frac{\exp\left(\beta w^\top \widetilde{\phi}\right)}{\widetilde{Z}(w)} P(w), \widetilde{Z}(w) = \int_{\widetilde{w}} \exp\left(\beta w^\top \widetilde{\phi}\right) d\widetilde{w}. \tag{2}$$

**Bayesian inverse reinforcement learning to approximate** $\widetilde{Z}$**.** During inference, the normalizing constant serves a calibration purpose: it computes how good the behavior produced by all proxy rewards in that MDP would be with respect to the true reward. Reward functions which increase the reward for *all* trajectories are not preferred in the inference. This creates an invariance to linear shifts in the feature encoding. If we were to change the MDP by shifting features by some vector $\phi_0$, $\phi \leftarrow \phi + \phi_0$, the posterior over $w$ would remain the same.

We can achieve a similar calibration and maintain the same property by directly integrating over the possible trajectories in the MDP:

$$Z(w) = \left(\int_{\xi} \exp(w^\top \phi(\xi)) d\xi\right)^{\beta} ; \quad \hat{P}(w|\widetilde{w}) \propto \frac{\exp\left(\beta w^\top \widetilde{\phi}\right)}{Z(w)} \tag{3}$$

---

[*]OpenAI, International Computer Science Institute (ICSI)

**Proposition 1.** *The posterior distribution that the IRD model induces on $w^*$ (i.e., Equation 2) and the posterior distribution induced by IRL (i.e., Equation 3) are invariant to linear translations of the features in the training MDP.*

*Proof.* First, we observe that this shift does not change the behavior of the planning agent due to linearity of the Bellman backup operation, i.e., $\widetilde{\phi}' = \widetilde{\phi} + \phi_0$. In Equation 2 linearity of expectation allows us to pull $\phi_0$ out of the expectation to compute $\widetilde{\phi}$:

$$\frac{\exp\left(\beta w^\top \widetilde{\phi}'\right)}{\int_{\widetilde{w}} \exp\left(\beta w^\top \widetilde{\phi}'\right) d\widetilde{w}} = \frac{\exp\left(\beta w^\top \phi_0\right) \exp\left(\beta w^\top \widetilde{\phi}\right)}{\int_{\widetilde{w}} \exp\left(\beta w^\top \phi_0\right) \exp\left(\beta w^\top \widetilde{\phi}\right) d\widetilde{w}} \tag{4}$$

$$= \frac{\exp\left(\beta w^\top \widetilde{\phi}\right)}{\int_{\widetilde{w}} \exp\left(\beta w^\top \widetilde{\phi}\right) d\widetilde{w}} \tag{5}$$

This shows that Equation 2 is invariant to constant shifts in the feature function. The same argument applies to Equation 3. □

## 2 Risk Averse Trajectory Optimization

Our overall strategy is to implement a system that 'knows-what-it-knows' about reward evaluations. So far we have considered the problem of computing the robot's uncertainty about reward evaluations. We have not considered the problem of using that reward uncertainty. Here we describe several approaches and highlight an important nuance of planning under a distribution over utility evaluations.

The most straightforward option is to maximize expected reward under this distribution. However, this will ignore the reward uncertainty we have worked so hard to achieve: planning in expectation under a distribution over utility is equivalent to planning with the mean of that distribution. Instead we will plan in a risk averse fashion that penalizes trajectories which have high variance over their utility. Of course, risk averse planning is a rich field with a variety of approaches (Markowitz, 1968; Rockafellar & Uryasev, 2000; Tamar et al., 2015). In future work, we intended to explore a variety of risk averse planning methods and evaluate their relative pros and cons for our application.

In this work, we will take a simple approach: given a set of weights $\{w_i\}$ sampled from our posterior $P(w|\widetilde{w}, \widetilde{M})$, we will have the agent compute a trajectory that maximizes reward under the *worst case* $w_i$ in our set. We can do this in one of two ways.

**Trajectory-wide reward.** Given a set of weights $\{w_i\}$ sampled from our posterior $P(w|\widetilde{w})$, we will have the agent compute a trajectory that maximizes reward under the *worst case* $w_i$ in our set:

$$\xi^* = \underset{\xi}{\operatorname{argmax}} \min_{w \in \{w_i\}} w^\top \phi(\xi). \tag{6}$$

This planning problem is no longer isomorphic to an MDP, as the reward may not decompose per state. Trajectory optimization in this case can be done via the linear programming approach described in Syed et al. (2008).

**Time-step independent reward.** An alternative is to take minimum over weights on a per state basis:

$$\xi^* = \underset{\xi}{\operatorname{argmax}} \sum_{s_t \in \xi} \min_{w \in \{w_i\}} w^\top \phi(s_t). \tag{7}$$

This is more conservative, because it allows the minimizer to pick a different reward for each time step.

Directly applying this approach, however, may lead to poor results. The reason is that, unlike maximizing expected reward, this planning approach will be sensitive to the particular feature encoding used. In maximizing expected reward, shifting all feature by a constant vector $\phi_0$ will not change the optimal trajectory. *The same is no longer true for a risk averse approach.*

Figure 1: **Left:** We avoid side effects and reward hacking by computing a posterior distribution over reward function and then find a trajectory that performs well under the worst case reward function. This illustrates the impact of selecting this worst case independently per time step or once for the entire trajectory. Taking the minimum per time step increases robustness to the approximate inference algorithms used because we only need one particle in our sample posterior to capture the worst case for each grid cell type. For the full trajectory, we need a single particle to have inferred a worst case for *every* grid cell type at once. **Right:**The impact of changing the offsets $c_i$. "Initial State" fixes the value of the start state to be 0. "Training Feature Counts" sets an average feature value from the training MDP to be 0. "Log Z(w)" offsets each evaluation by the normalizing from the maximum entropy trajectory distribution. This means that the sum of rewards across a trajectory is the log probability of a trajectory.

For example, consider a choice between actions $a_1$ and $a_2$, with features $\phi_1$ and $\phi_2$ respectively. If we shift the features by a constant value $-\phi_2$ (i.e., set the feature values for the second action to 0), then, unless $a_1$ is preferred to $a_2$ for *every* weight in the posterior, the agent will always select the second action. The zero values of feature encodings are typically arbitrary, so this is clearly undesireable behavior.

Intuitively, this is because rewards are not absolute, they are relative. Rewards need a reference point. We thus need to compare the reward $w^T \phi(\xi)$ to *something*: to the reward of some reference features $c_i$. We will study three approaches: comparing reward to the initial state, to the training feature counts, and to the expected reward across any trajectory.

**Comparing to initial state.** One straightforward approach is to take a particular state or trajectory and enforce that it has the same evaluation across each $w_i$. For example, we can enforce that the features for the initial state state is the 0 vector. This has the desirable property that the agent will remain in place (or try to) when there is very high variance in the reward estimates (i.e., the solution to IRD gives little information about the current optimal trajectory).

**Comparing to training feature counts.** An third option is to use the expected features $\widetilde{\phi}$ as the feature offset. In the case, the agent will default to trying to match the features that it would have observed maximizing $\widetilde{w}$ in $\widetilde{M}$.

**Comparing to other trajectories.** An alternative is to define $c_i$ as the log of the normalizing constant for the maximum entropy trajectory distribution:

$$c_i = \log \int_\xi exp(w_i^\top \phi(\xi)) d\xi. \tag{8}$$

With this choice of $c_i$, we have that $w_i^\top \phi(\xi) - c_i = \log P(\xi|w_i)$. Thus, this approach will select trajectories that compare relatively well to the options under all $w_i$. Loosely speaking, we can think of it as controlling for the total amount of reward available in the MDP.

**Evaluation.** Before running the full experiment, we did an initial internal comparison to find the best-performing planning method. We did a full factorial across the factors with the side effect feature encoding and the reward hacking feature encoding.

We found that the biggest overall change came from the **min-granularity** feature. A bar plot is shown in Figure 1 (Left). Independently minimizing per time step was substantially more robust. We hypothesize that this is a downstream effect of the approximate inference used. We sample from our belief to obtain a particle representation of the posterior. Independently minimizing means that we

need a single particle to capture the worst case for each grid cell type. Performing this minimization across the full trajectory means that a single particle has to faithfully represent the worst case for *every* grid cell type.

We also saw substantial differences with respect to the **reward-baseline** factor. Figure 1 (Right) shows a bar plot of the results. In this case, setting the common comparison point to be the average feature counts from the training MDP performed best. We believe this is because of the similarity between the train and test scenarios: although there is a new grid cell present, it is still usually possible to find a trajectory that is similar to those available in the training MDP. We hypothesize that correctly making this decision will be situational and we look forward to exploring this in future work.