[Reviews · NeurIPS 2017]

Reviewer 1



This paper proposes a new problem, whereby an agent must in effect select an optimal reward function, from observing a proxy (incomplete) reward function and a set of trajectories. This is a very interesting extension of the well-studied inverse reinforcement learning, which offers useful new perspectives on the value alignment problem. The proposed problem is clearly defined, relevant literature is discussed. The formulation is quite elegant, and it is a pleasant surprise to see the interpretation that comes out of the bayesian IRL case, whereby the proxy reward reduces to providing a set of corresponding demonstrations. I liked both the problem formulation and the proposed approximations to the IRD posterior. The main aspect of the paper that could be improved is the experiments. Currently, there are simple toy MDPs that serve as proof of concept. For illustrative purposes, they are suitable and experiments are correctly executed. But I would be interested in seeing the method used for a real-world MDP, where reward is obtained from a human. A few minor points that could be improved: - Notation in Definition 3 gets confusing; the different fonts used for the reward (proxy reward vs space of possible reward) are hard to parse visually. - It may be interesting to relate the proxy reward to a notion of "necessary but not sufficient" reward information. - Sec.4: It would be useful to discuss the complexity of the approximations discussed. - Fig.4: Add y-axis label. - References: Some inconsistencies in style and missing information, e.g. where was Singh et al published? In summary, the paper is novel, thought-provoking, and tackles an important problem. I recommend acceptance.

Reviewer 2



The authors develop "Inverse Reward Design" which is intended to help mitigate unintended consequences when designing reward functions that will guide agent behaviour. This is a highly relevant topic for the NIPS audience and a very interesting idea. The main potential contribution of this paper would be the clarity of thought it could bring to how issues of invariance can speak to the IRL problem. However, I find that the current presentation to be unclear at many points, particularly through the development of the main ideas. I have done my best to detail those points below. My main concern is that the presentation leading up to and including 4.2 is very confused. The authors define P(\tilde w | w^*). (They always condition on the same \tilde M so I am dropping that.) At l.148, the authors state that through Bayes' rule, get P(w^* | \tilde w) \propto P(\tilde w | w^*) P(w^*) In the text, the display just after l.148 is where things start to get confused, by the use of P(w) to mean P(w^* = w) The full Bayes' rule statement is of course P(w^* | \tilde w) = P(\tilde w | w^*) P(w^*) / P(\tilde w), or P(w^* | \tilde w) = P(\tilde w | w^*) P(w^*) / \int_\w^* P(\tilde w | w^*) P(w^*) dw^* I am confused because 1) the authors don't address the prior P(w^*) and 2) they state that the normalizing constant integrates "over the space of possible proxy rewards" and talk about how this is intractable "if \tilde{w} lies in an infinite or large finite set." Since w^* and \tilde{w} live in the same space (I believe) this is technically correct, but a very strange statement, since the integral in question is over w^*. I am concerned that there is substantial confusion here at the core idea of the whole paper. I suspect that this has resulted just from notational confusion, which often arises when shorthand [P(w^*)] and longhand [P(w^* = w)] notation is used for probabilities. I also suspect that whatever was implemented for the experiments was probably the right thing. However, I cannot assess this in the current state of the paper. Finally, I would suggest that the authors address two questions that I think are natural in this setting. 1) What do we do about the prior P(w^*)? (Obviously there won't be a universal answer for this, but I imagine the authors have some thoughts.) 2) Below are additional comments: l.26 - The looping example isn't given in enough detail to help the reader. The example should stand alone from the cited paper. Figure 1 - I understand the podium in the figure on the right, but the meaning of the numbers is not clear. E.g. is the first number supposed to convey [2][0] or twenty? l.50 - "Extracting the true reward is..." - Do you mean "We define..." Figure 2: uknown l.134 - I think somewhere *much* earlier it needs to be made clear that this is a feature-based reward function that generalizes over different kinds of states. l.138 - \xi and \phi(\xi) are not precisely defined. I don't know what it means for an agent to "select trajectory \xi" l.145 - Notation in (1) and (2) is confusing. I find conditioning on \xi drawn from ... inside the [] strange. Subscript the expectation with it instead? l.148 - P(w) should be P(w|\tilde{M}) unless you make additional assumptions. (true also in (3)) On the other hand, since as far as I can tell everything in the paper conditions on \tilde{M}, you may as well just drop it. l.149 - Probabilities are normalized. If you're going to talk about the normalizing constant, write it explicitly rather than using \propto. l.150 - Is this the same w? Looks different. l.151 - I think you want = rather than \propto. Something is wrong in the integral in (3); check the variable of integration. There are \bar\phi and \tilde\phi and it's not clear what is happening. l.158 - "constrained to lie on a hypercube" - Just say this in math so that it's precise. l.166 - Can't (4) just go from 1 to N? l.228 - I wouldn't open a figure caption with "Our proof-of-concept domains make the unrealistic assumption." Furthermore, I have no idea how the figure caption relates to the figure. === UPDATE The authors have clarified section 4 and I understand much better what they were after. I have updated my score accordingly.

Reviewer 3



The paper "Inverse Reward Design" discusses Markov decision processes (MDPs) where the reward function is designed by an expert who makes assumptions which may be specific to the dynamics of the original MDP for which the reward function was specified. However, the reward function may not directly generalize to new MDPs since the expert may not be able to take everything into account when designing the original reward function. The paper defines the inverse reward design (IRD) problem as finding a probability distribution over the true reward function for a new MDP given a reward function for the original MDP. The investigated problem is interesting and has real-world relevance. The proposed computational approaches work in the tested problems. The paper is well written. For computing the probability distribution over the true reward function, the paper assumes a maximum entropy distribution over trajectories given the expert's reward function and the original MDP. Using this assumption the distribution over the weights of the true reward function can be defined. To make computations tractable the paper presents three different approximation schemes and evaluates the schemes in problems designed specifically to investigate the IRD problem. The results show that the IRD approaches perform better compared to using the original reward function when using a selected MDP planning approach which avoids risks. I would like the authors to answer following question: - In general, the solution to IRL is non-unique and therefore often a maximum entropy assumption is used. However, here we have information about possible solutions since we are given a reward function designed by the expert. Could we here in principle compute a solution to the IRD problem without the maximum entropy (or similar) assumption? If not, a practical example would be nice? DETAILS For the example in Figure 1 Left a more formal definition of the example could clarify the picture? The example in Figure 1 Right is confusing. There are two problems: 1) A clear definition of reward hacking is missing 2) The picture can mean many different things for different readers. Now the picture looks like that the designed reward should actually work but why it does not work is unclear. The caption talks about pathological trajectories and not winning the race but the picture does not address these points using concrete examples. Maybe a top down picture of a race track where the agent gets high reward for undesired behavior such as cutting corners or driving over people, or similar, could help? RELATED WORK In the IRD problem, the expert is required to define a reward function. How does IRD relate to other kinds of goal specifications (e.g. logic constraints)? Could the approach be extended to these kind of settings? EVALUATION The text "In the proof-of-concept experiments, we selected the proxy reward function uniformly at random. With high-dimensional features, most random reward functions are uninteresting. Instead, we sample 1000 examples of the grid cell types present in the training MDP (grass, dirt, and target) and do a linear regression on to target reward values." should be clarified. Now, the paper says that the reward function is selected uniformly at random but that is a bad idea and instead 1000 examples are sampled? Figure 4: What do the error bars denote? Units for the y-axis should be added. How is literal-optimizer represented in Figure 4? LANGUAGE "we then deploy it to someone’s home" -> "we then deploy the robot to someone’s home" "the bird is alive so they only bother to specify" Who is they? "as this is an easier reward function to learn from." easier than what? "uknown feature." -> "unknown feature." "might go to the these new cells" -> "might go to these new cells" "subsample the the space of proxy" -> "subsample the space of proxy"